# Sodium-Glucose Cotransporter-2 Inhibitors in Liver Cirrhosis: A Systematic Review of Their Role in Ascites Management, Slowing Disease Progression, and Safety

**DOI:** 10.3390/ijms26104781

**Published:** 2025-05-16

**Authors:** Sudheer Dhoop, Sami Ghazaleh, Luke Roberts, Mohammed Shehada, Manthanbhai Patel, Wade-Lee Smith, Sana Rabeeah, Bisher Sawaf, Priya Vadehra, Benjamin Hart, Mona Hassan

**Affiliations:** 1Department of Internal Medicine, The University of Toledo, 3333 Glendale Ave., Toledo, OH 43614, USAbisher.sawaf@utoledo.edu (B.S.); 2Division of Gastroenterology and Hepatology, The University of Toledo, 3333 Glendale Ave., Toledo, OH 43614, USA; 3College of Medicine and Life Sciences, The University of Toledo, 3000 Arlington Ave., Toledo, OH 43614, USA; 4University Library, The University of Toledo, 2801 W. Bancroft St., Toledo, OH 43606, USA; 5Department of Biological Sciences, Wayne State University, 4841 Cass Ave., Detroit, MI 48201, USA; hl0790@wayne.edu

**Keywords:** SGLT2I, refractory ascites, hepatic decompensation, hypotension, infection, AKI, electrolyte abnormalities

## Abstract

Sodium-glucose cotransporter-2 inhibitors (SGLT2Is) are widely used for type 2 diabetes mellitus (T2DM), conferring cardiovascular and renal benefits with evidence supporting their role in metabolic-associated steatotic liver disease (MASLD), the fastest rising etiology for liver cirrhosis. Our study collects and synthesizes all available data on SGLT2I use in liver cirrhosis to summarize their potential benefits and risks. We systematically reviewed the literature on SGLT2I use in adults with cirrhosis, focusing on 6 outcome domains, including ascites reduction, disease progression, hemodynamics, acute kidney injury (AKI), electrolyte abnormalities, and infection risk. We identified 16 studies: compensated (*n* = 5), decompensated (*n* = 3), and refractory ascites (*n* = 8). All studies of decompensated cirrhosis (*n* = 11) reported ascites reduction. Most studies (7 of 9) indicated SGLT2Is slowed disease progression by reducing clinical decompensation (*n* = 4) or improving laboratory markers (*n* = 3). A minority of studies revealed safety concerns with 2 of 9 studies showing evidence of hemodynamic instability and acute kidney injury (AKI), 2 out of 13 for electrolyte abnormalities, and 2 out of 5 for infection risk. Current evidence strongly supports SGLT2Is for refractory ascites management and suggests potential benefits in slowing progression across cirrhosis severities. Longer-term prospective trials in patients with non-refractory decompensated cirrhosis and real-world safety data are essential to clarify and potentially expand the role of SGLT2Is in cirrhosis management.

## 1. Introduction

Sodium-Glucose Cotransporter-2 Inhibitors (SGLT2Is) block glucose and sodium reabsorption at the renal proximal tubule, triggering natriuresis and osmotic diuresis [1]. They were approved by the Food and Drug Administration (FDA) for glycemic control in Type 2 Diabetes Mellitus (T2DM) in 2013 [1]. Their use has doubled from 2015 to 2020, owing to their preventative benefits of cardiovascular and renal disease independent of T2DM [2,3,4]. Patients with Metabolic-Associated Steatotic Liver Disease (MASLD), formerly Non-Alcoholic Fatty Liver Disease (NAFLD), have cardiometabolic risk factors, including diabetes, so they are often prescribed SGLT2Is [5]. Strong clinical evidence reveals that SGLT2Is lead to improved liver fibrosis, liver enzyme levels, and inflammatory markers in MASLD [6,7]. Therefore, the American Association of Clinical Endocrinology recommends their use as an adjunct treatment for T2DM in MASLD [1]. In the U.S., MASLD is the leading cause of liver cirrhosis, with MASLD being the most rapidly rising indication for liver transplantation [8]. Given the rising prevalence of MASLD-related cirrhosis alongside the expanding use of SGLT2Is in diabetes, a major risk factor for MASLD, the number of patients with cirrhosis receiving SGLT2Is is likely increasing. Therefore, it is critical to assess how SGLT2I use affects complications of liver cirrhosis, such as ascites, hepatic decompensation, as well as their safety in patients with liver cirrhosis. Data suggests that SGLT2I use may help reduce fibrosis in advanced MASLD.

In a study of patients with severe hepatic fibrosis with liver stiffness measurements of >10 kPa, there was an association between SGLT2I use and reduced liver stiffness measurements (LSMs) on vibration-controlled transient elastography (VCTE) [9]. While all these patients may not have had confirmed cirrhosis, they resembled patients with compensated cirrhosis, defined as a liver stiffness of >12.0 kPa in MASLD [10]. Therefore, SGLT2Is may have the potential to improve fibrosis in the early stages of cirrhosis and thus may have a role in compensated cirrhosis, as suggested by a review of SGLT2I use in cirrhosis [11]. A cohort study corroborates this as it found SGLT2Is were associated with reduced incidence of esophageal varices, which are driven by liver stiffness through portal hypertension [12]. Trials have found SGLT2Is slow architectural changes that drive portal hypertension in carbon tetrachloride-induced cirrhosis in rats and reduce clinically significant portal hypertension (CSPH) alongside zibotentan in all etiologies of compensated cirrhosis patients [13,14]. Additionally, a cohort study found genetic proxies for SGLT2 inhibition were associated with lower hazards of hepatic decompensation, providing further evidence that SGLT2Is may improve liver function in cirrhosis [15]. In terms of decompensated cirrhosis, a compilation of case reports outlined in reviews have also shown that SGLT2Is have a role in patients with refractory ascites in cirrhosis of various etiologies through their diuretic effect when used synergistically with loop diuretics and mineralocorticoid receptor antagonists which are the standard of care diuretics (SoCD) for cirrhosis related ascites [16,17,18,19,20,21].

Major side effects of SGLT2Is include genital mycotic infections, including Fournier’s gangrene or urinary tract infection, including urosepsis and pyelonephritis, risk of limb amputation, euglycemic diabetic ketoacidosis, dyslipidemia, volume contraction leading to AKI, and hypokalemia [1]. Many of these risks are of heightened concern in cirrhosis, particularly decompensated cirrhosis, an advanced disease stage where severe hepatic impairment leads to abdominal ascites, variceal bleeding, hepatic encephalopathy (HE), hepatorenal syndrome (HRS), and jaundice [22]. In decompensated disease, portal hypertension drives splanchnic vasodilation, leading to systemic hypotension, which reduces renal perfusion [23]. Loop diuretics can trigger volume depletion and hypotension, leading to acute kidney injury (AKI) or hepatorenal syndrome–acute kidney injury (HRS–AKI), so the same may hold true for the diuretic effect of SGLT2Is combined with SoCDs [24]. This combined effect may also cause electrolyte disturbances with sodium and potassium, which cirrhotic patients are already prone to [25]. Moreover, SGLT2Is have been shown to increase mycotic genital infections and potentially urinary tract infections with higher doses, which is a concern in cirrhotic individuals with impaired innate immunity [26,27,28]. Finally, SGLT2Is are known to be associated with diabetic ketoacidosis (DKA) with elevated or normal (euglycemic) glucose levels in T2DM [29,30]. Patients with cirrhosis have been shown to have worse inpatient outcomes in DKA [31]. Additionally, SGLT2I triggering DKA in T2DM in alcoholic cirrhosis has been documented and may be a concern, as active alcohol use and starvation associated with alcoholic cirrhosis can further contribute to ketoacidosis [32,33].

Since the last review on SGLT2I use in cirrhosis in early 2024 [11], this has become an emerging area of research with an abundance of studies published afterwards, including the first two randomized controlled trials, studying SGLT2Is in various forms of liver cirrhosis for refractory ascites management [34,35]. Therefore, the purpose of this review is to systematically identify all studies and reports where SGLT2Is are used in liver cirrhosis to provide a summary on their therapeutic role in ascites, potential to slow disease progression, and safety, as well as identify gaps to direct future research.

## 2. Materials and Methods

A comprehensive search strategy to identify studies involving patients with cirrhosis exposed to SGLT2Is was developed in Embase (Embase.com, Elsevier N.Y., USA) by an experienced health science librarian (W.-L.S.) using truncated keywords, phrases, and subject headings (see Appendix A); it was translated to MEDLINE (PubMed platform, National Center for Biotechnology Information, National Library of Medicine), Cochrane Central Register of Controlled Trials, Web of Science Core Collection, Korean Citation Index (Web of Science platform, Clarivate), and Global Index Medicus. The initial search was conducted on 28 October 2024 and updated on 31 January 2025 (Appendix A). No publication date, country, or language exclusions were applied. All results were exported to EndNote 21 citation management software (Clarivate, Philadelphia, PA, USA).

To screen studies, duplicates were removed by successive reiterations of EndNote’s duplicate detection algorithms and manual inspection. Two authors (S.D., L.R.) reviewed the records. Both comparative and non-comparative studies (including case reports) were considered. Excluded records included duplicates not removed by software, reviews, studies involving animals or children, preprints, and studies where not all patients had cirrhosis, not all patients in a study arm were given SGLT2Is, or where SGLT2I was combined with another experimental treatment. Studies that assessed SGLT2I effect on hepatic fibrosis alone were excluded unless all patients were noted to have at least compensated cirrhosis. Conference abstracts were considered, but excluded if there was concern, the study populations overlapped with another abstract. Another author (S.G.) was available to discuss disputes in study inclusion. The study selection process followed the flow diagram outlined in Figure 1 as recommended by the Preferred Reporting Items for Systematic reviews and Meta-Analyses (PRISMA) 2020 statement [36].

The baseline study population characteristics, interventions, controls, primary or significant outcomes, and time periods (PICOT) were extracted and outlined in Table 1 for comparative studies and Table 2 for non-comparative studies. We systematically reviewed each study to assess six outcome domains with SGLT2I use based on the evidence mentioned above. These outcome domains were ascites reduction, slowed cirrhosis progression, including improved lab parameters or reduced hepatic decompensations, incidence of hemodynamic instability, incidence of AKI, effect on electrolyte abnormalities, and incidence of infection. Studies were categorized into subgroups based on the predominant cirrhosis severity: compensated cirrhosis, decompensated cirrhosis, and decompensated cirrhosis with refractory ascites. Of note, studies with decompensated cirrhosis not specifying refractory ascites status were placed in the decompensated subgroup.

Ascites reduction was considered as any objective evidence of reduced ascites through reduced diuretic dose and/or paracentesis or imaging evidence. Slowed cirrhosis progression was defined as (1) Improvement in model for End-Stage Liver Disease with Sodium (MELD-Na) (2) improvements in any individual synthetic or metabolic parameters including INR, Albumin, Bilirubin, or (3) decreased incidence of decompensations including ascites onset (in compensated cirrhosis), HE or variceal bleeding. Hemodynamic instability included hypotension, tachycardia, or pressor requirement, which was reported as defined by each study. Electrolyte abnormalities included the incidence of sodium, potassium, and acid/base abnormalities as defined by each study. Changes in these electrolyte values were also noted. Incidence of all infections was grouped together, as were various etiologies for AKIs, and were extracted as reported/defined by individual studies. For comparative studies, we deemed a positive association with SGLT2Is if there was a statistical difference from the control. A single incident or a consistent increase or decrease in labs was considered a positive association for non-comparative studies. The quality assessment of these studies is summarized in Table 3. Results for each domain were tallied (Table 4) without pooling data to determine effect sizes and illustrated in Figure 2.

Quality assessment of studies was performed using the Cochrane Risk of Bias 2.0 (RoB 2.0) Tool for Randomized Controlled Trials (RCTs), the Risk Of Bias In Non-randomized Studies of Interventions (ROBINS-I) tool for nonrandomized studies, and the National Institute of Health Quality Assessment Tool (NIHQAT) for Before–After (Pre-Post) Studies and for case series as outlined in Table 4 [47,48,49]. No assessment was performed on individual case studies. The PRISMA checklist was utilized to include all elements pertinent to a systematic review outlined in Appendix A [36]. While several studies measured similar outcomes, pooling of the data was not useful due to heterogeneity between studies, given the differences in study design, populations, and time periods reported, so meta-analysis was deferred [50]. Instead, in addition to tallying results of each domain, the studies included were critically appraised in the discussion to weigh the certainty of the evidence.

## 3. Results

### Summary of Results

Our review process (Figure 1) yielded a total of 16 studies: 15 peer-reviewed manuscripts [16,17,18,32,34,35,37,38,40,41,42,43,44,45,46] and one conference abstract [39]. Two additional conference abstracts were excluded, including one assessing SGLT2Is combined with an- other experimental agent, zibotentan, and another because it queried the same database as the included abstract [14,51]. Additionally, one trial otherwise met eligibility criteria but was pre-print, so it was excluded [52]. Table 1 and Table 2 summarize the characteristics of the included studies, which comprise six comparative [34,35,37,38,39,40] and ten non-comparative studies [16,17,18,32,41,42,43,44,45,46]. Of these, five focused on compensated cirrhosis [32,37,38,41,42] and ten on decompensated cirrhosis [16,17,18,34,35,39,40,44,45,46], with most decompensated studies (*n* = 8) specifically evaluating patients with refractory ascites [16,17,18,34,35,43,44,46]. One focused on a 50–50 mix of compensated and decompensated cirrhosis [43].

Eleven studies reported the role of SGLT2 inhibitors in ascites management, and all documented a reduction in ascites [16,17,18,34,35,39,40,43,44,45,46]. Nine studies reported hepatic parameters beyond ascites reduction, with seven demonstrating potential benefits—two in compensated cirrhosis, three in decompensated cirrhosis, and two in patients with refractory ascites [17,34,35,37,38,39,40,42,45]. Nine studies assessed the risk of hemodynamic instability [16,17,34,35,40,42,43,44,46]; one of these reported an incidence of it in a single-arm study in compensated cirrhosis [43] and another in refractory cirrhosis [46]. Similarly, among nine studies examining the risk of AKI [16,17,34,35,41,42,43,44,46], two identified an increased incidence in patients with refractory ascites [35,46]. Thirteen studies [16,17,18,32,34,35,40,41,42,43,44,45,46] evaluated electrolyte and acid–base disturbances, with six reporting improvements in hyponatremia [16,17,18,42,44,45], one noting worsening hyponatremia [34], and one case reporting ketoacidosis associated with SGLT2I exposure [32]. Finally, five studies assessed infection risk [34,35,40,41,44] with one study showing an increased risk of total infections or incidence of urinary tract infection (UTI) in refractory ascites [35] and another reporting the incidence of genital infections in patients with compensated cirrhosis [41].

Quality assessment revealed the RCTs had low to some risk of bias due to withdrawals and blinding [34,35]. Cohort studies quality varied from low, moderate, high risk of bias, with concerns due to incomplete account of confounders, suboptimal control groups, and selective outcome reporting [37,40,41,46]. All non-comparative studies lacked blinding of outcome assessment and were of single-center design in low sample size groups.

## 4. Discussion

### 4.1. Overview

SGLT2Is are increasingly prescribed for patients with cardiovascular risk factors. These risk factors often overlap with MASLD, which is a rapidly growing cause of cirrhosis, and frequently coexists with other cirrhosis etiologies. Therefore, understanding the effects of SGLT2I use in cirrhotic populations is increasingly relevant for future clinical practice. Our systematic review examined the current literature to identify key themes and knowledge gaps. We found SGLT2Is to be consistently effective in reducing refractory ascites for up to six months. However, evidence for their benefit in decompensated cirrhosis (without refractory ascites) or compensated cirrhosis is limited and weak. Our review does have some limitations. Ten studies lacked a control group. This hinders the assessment of SGLT2I effects compared to clinical alternatives. Furthermore, these studies often presented limited data, in some instances relying on case reports. Beyond controlled trials focused on SGLT2I efficacy in ascites management, the remaining studies evaluated SGLT2I exposure. Consequently, some of these studies may have selection bias. This bias arises from confounding factors, which limit the comparability of different patient groups. Finally, despite a similar intervention across studies and our ability to categorize outcomes into six domains, heterogeneity existed. Study populations, control groups, outcome measurement methods, and follow-up durations varied based on individual research questions. This variability reduces the direct comparability between studies. Therefore, the following discussion of outcome domains will critically appraise our review findings.

### 4.2. Ascites Control

Ascites is the most frequent complication leading to a diagnosis of decompensated cirrhosis [53]. Ascites has significant clinical consequences. These include patient discomfort, respiratory compromise, and an increased risk of infection. Therefore, ascites control is crucial, with diuretics as the first-line treatment. Numerous studies in our review indicate that SGLT2Is can help reduce the ascites burden. The strongest evidence supports their use in refractory ascites. Specifically, SGLT2Is improve ascites that has not responded to SoCD at maximal doses (diuretic-resistant ascites). They are also beneficial when SoCDs are contraindicated due to side effects (diuretic-intractable ascites). This aligns with cardiology research. In congestive heart failure (CHF), SGLT2Is have shown some effectiveness in reducing loop diuretic doses and enhancing diuresis in diuretic-resistant congestive heart failure [54]. Our review identified two small RCTs (Bakosh 2024 and Singh 2024) in cirrhosis patients with refractory ascites. These RCTs found that adding SGLT2Is to SoCDs, or using SGLT2Is when SoCDs were contraindicated, significantly reduced ascites. They also observed decreased loop diuretic dosages and a reduced need for large volume paracentesis (LVP) [34,35]. The latter is critical as LVP is an invasive, costly procedure with risks. These risks include precipitating fluid shifts and increasing the risk of AKI or HRS-AKI [55]. Notably, Bakosh 2024 further categorized refractory ascites. This categorization included diuretic-resistant ascites (ascites despite maximal SoCD doses) and diuretic-intractable ascites (maximal SoCD doses not achievable) [34]. The diuretic-resistant group required more LVPs than the intractable group. This suggests SGLT2Is may enhance loop diuretics when maximal doses are not tolerated. However, they may be less effective at enhancing maximal SoCD doses. A pilot study by Kalambokis 2024 in refractory ascites showed that SGLT2Is increased natriuresis. This was reflected by a tripling of 24 h sodium excretion at 1 month, sustained over 3 months, and reduced excess water retention [44]. The Singh 2024 RCT corroborated this finding. It noted significantly increased 24 h sodium excretion in the SGLT2I group compared to SoCD alone, lasting up to 6 months [35]. However, urine volume did not significantly increase. This shows their natriuretic effect is more prolonged than their overall diuretic effect. This is interestingly different from SGLT2Is in CHF, where the natriuretic effect is short-lived [54]. Like CHF, SGLT2Is in cirrhosis reduced markers of Renin–Angiotensin–Aldosterone System (RAAS) activity, such as plasma renin and aldosterone [52]. This reduction would decrease free water retention in primarily interstitial spaces and improve ascites control [44]. Further research is needed on the efficacy of SGLT2Is in cirrhosis past 6 months with refractory ascites, as when used with SoCDs, they may carry a more sustained diuretic effect than they do in CHF. Additionally, the diuretic mechanism of SGLT2Is appears independent of significant glucose excretion, as one study reported no correlation between urinary glucose levels and markers of water–sodium retention [52]. This suggests that SGLT2Is could effectively manage ascites irrespective of diabetes status, aligning with findings from our review, where only 5–50% of participants in the aforementioned RCTs had T2DM.

Data on SGLT2I use in decompensated cirrhosis for non-refractory ascites are limited. Seif El-Din 2024 conducted a prospective study. It compared SGLT2I administration (without insulin) to subcutaneous insulin primarily in decompensated cirrhotic patients. The study found insulin users required higher diuretic doses (41% vs. 25%, *p* < 0.01) [40]. Although a primary outcome was not specified, Seif El-Din 2024 concluded SGLT2Is were effective for ascites management. However, this conclusion is questionable. A significant 20% of insulin users reduced or stopped diuretics, compared to 0% in the SGLT2I-only group. This observation contradicts the claim that SGLT2Is effectively controlled ascites in this study. Furthermore, the study lacked randomization and blinding. This likely introduced a confounding variable, such as differences in baseline medications. Using insulin as a comparator is also not ideal. Insulin may increase adrenergic activity, potentially worsening portal hypertension and edema [56,57]. A large database study by Ayoub 2024, (published as a conference abstract), retrospectively analyzed the TriNetX Research Network. This study compared propensity-matched cohorts in decompensated cirrhosis. One group received SGLT2Is in addition to SoCD, and the other received SoCD alone. Ayoub 2024 reported significant reductions in paracentesis in the SGLT2I group at one, three, and six months [39]. However, this database study has limitations. The lack of detailed data makes it difficult to distinguish between SGLT2I prescriptions for refractory ascites and exposure in decompensated cirrhosis due to cardiovascular risk. Lastly, Seidita 2024 published a case series of four patients as a preview of an ongoing trial of SGLT2Is in decompensated cirrhosis [45]. Notably, most patients in this series (three out of four) started SGLT2Is for increased cardiovascular risk, not refractory ascites. Despite this, all patients showed a substantial reduction in ascites grade. Three patients even achieved complete ascites resolution at six months. Therefore, preliminary evidence suggests SGLT2Is may have a role in decompensated, non-refractory ascites.

### 4.3. Slowed Disease Progression

Several studies have evaluated the effect of SGLT2 inhibitors (SGLT2Is) beyond ascites reduction, including prevention of decompensation and reduction of inflammation. Saffo 2021 conducted a well-designed retrospective cohort study. This study focused on a homogeneous U.S. male veteran population with compensated cirrhosis, mainly MASLD. It assessed ascites incidence at 3 years. The study used propensity matching for liver function, fibrosis severity, demographics, and medication use [37]. Saffo 2021 found that SGLT2I exposure was associated with a decrease in ascites incidence at 3 years. However, this decrease was statistically insignificant compared to Dipeptidyl Peptidase-4 Inhibitors (DPP4is), another oral diabetic medication without known edemogenic effects. Huynh 2023 performed another retrospective cohort study using similar statistical methods and the U.S. TriNetX Research Network data. This study reported a significant decrease in overall decompensations (ascites, varices, and HE) as a secondary outcome. However, Huynh 2023 had a potential confounder with the metformin-only group having significantly higher MELD-Na scores at baseline. Sharma 2023 published a case series in India on compensated cirrhosis, with 85% having MASLD. This series reported a 3.4 kg weight loss and reduced transaminases [42]. Decreased transaminases and weight loss could indicate reduced liver inflammation and adipose tissue, like findings in non-cirrhotic MASLD [6,7]. Further prospective studies are needed. These studies should evaluate whether the benefits of adding SGLT2Is in this population outweigh the risks.

There is also some evidence suggesting the benefit of SGLT2Is on intrinsic liver function in decompensated cirrhosis to slow or reverse disease progression. Ayoub 2024 reported a significant association between SGLT2I use and fewer hepatic decompensation events for up to 6 months [39]. However, propensity-matched baseline characteristics of key confounders, like MELD-Na scores, were not provided. This raises concern that SGLT2Is might have been preferentially used in less critically ill patients. Supporting this concern, Seif-El Din 2024 compared SGLT2I use to insulin. While SGLT2Is were associated with lower variceal bleeding and HE, the insulin group had higher baseline MELD scores. Seidita 2024’s prospective case series in decompensated cirrhosis patients showed clinically significant reductions in ascites and Child–Pugh scores [45]. Additionally, one RCT in decompensated cirrhosis with refractory ascites indicated the SoCD group had a 3-point increase in MELD-Na. In contrast, the SGLT2I group’s MELD-Na remained stable. This may suggest SGLT2Is slow cirrhosis progression. However, liver prognostic scores might improve with SGLT2I use mainly due to ascites reduction. This can also improve serum sodium and reduce creatinine due to lean body mass reduction with SGLT2Is [58]. Other liver function markers, such as INR, albumin, and bilirubin, were unchanged or inconsistently improved. Overall, current evidence suggests SGLT2Is may reduce hepatic decompensation or slow liver cirrhosis progression for up to 6 months.

The basis for SGLT2Is role in slowing disease progression is likely due to their pleiotrophic effects on the liver (see Figure 2). They have been shown to reduce hepatic inflammation, evidenced by a reduction in serum ferritin and liver enzymes through a reduction in oxidative stress [6,7]. This reduction in repeated inflammation is likely to lead to antifibrotic effects demonstrated by reductions in liver stiffness, which likely contributes to reductions in portal hypertension, a driving force for variceal bleeding [9,12,13,22]. HE episodes may also be reduced through reductions in portal pressure and reduction in loop diuretics observed, which may reduce the incidence of hypokalemia, which can trigger HE. In terms of long-term benefits on liver architecture, a meta-analysis reveals an association with SGLT2I and HCC incidence in older populations with T2DM and/or MASLD, so they may also carry anti-neoplastic properties as they were associated with decreased risk of other gastrointestinal cancers [59]. This effect was not observed across all populations, and more prospective data is needed on this topic. Overall, our review demonstrates that there is mechanistic and real-world evidence to suggest SGLT2Is may slow the progression of advanced liver disease, which warrants further clinical trials.

### 4.4. Hemodynamic Effects

A potential concern with SGLT2Is is hypotension. Their diuretic effect might worsen pre-existing hypotension in cirrhosis, especially when combined with other diuretics. Two studies suggest this possibility. However, most evidence indicates no increased hypotension (or no significant difference) and even potential hemodynamic benefits. Both RCTs adding SGLT2Is to SoCDs and the Seif El-Din 2024 prospective study reported similar hypotension rates in both SGLT2I and control groups. Notably, the Bakosh 2024 RCT showed a trend towards lower Mean Arterial Pressure (MAP) in the SoCD group. One caveat is that in the Bakosh 2024 RCT, 10% of SGLT2I patients withdrew due to AKI. This withdrawal could have underestimated hypotension incidence and the extent of MAP reduction associated with SGLT2Is, so further investigation is warranted. Regarding hemodynamic benefits, Kalambokis 2024 primarily investigated SGLT2I effects on hemodynamic parameters. This study found improved cardiac function. Although there was a statistically significant 2-point MAP reduction, cardiac output normalized. This normalization suggests a reversal of hyperdynamic circulation, a key feature of cirrhotic cardiomyopathy (CCM), a cardiac dysfunction in cirrhosis [60]. Furthermore, Left Atrial Volume Index (LAVI) and E/e’ ratio, markers of diastolic dysfunction, normalized during empagliflozin treatment. These findings are consistent with SGLT2I efficacy in diastolic heart failure. Kalambokis 2024 was the first to demonstrate such efficacy in cirrhosis patients possibly with CCM. Regarding the MAP drop, another single-arm prospective study in compensated cirrhosis also showed a similar significant MAP reduction at 3 months [42]. However, this study’s 6-month outcomes showed MAP normalization.

These hemodynamic benefits align with widely cited data from heart failure with preserved ejection fraction (HFpEF), a condition that shares several pathophysiologic features with cirrhotic cardiomyopathy. In the PRESERVED-HF trial, dapagliflozin significantly improved both patient-reported outcomes and physical capacity [61]. These benefits were seen regardless of diabetes status. Additional trials, including DELIVER and EMPEROR-Preserved, have demonstrated reductions in heart failure hospitalizations and improved symptom burden among patients with left ventricular ejection fraction (LVEF) > 40% [62,63]. While cirrhotic cardiomyopathy is not explicitly represented in these studies, the overlapping mechanisms raise the possibility that SGLT2Is may offer similar cardiovascular benefits in cirrhotic populations.

### 4.5. Acute Kidney Injury Concerns

Beyond hypotension, the diuretic effect of SGLT2Is also raises concerns about AKI risk. With multiple mechanisms driving hypotension in cirrhosis, excessive volume depletion due to diuretics can reduce renal perfusion. This reduced perfusion may trigger AKI or HRS-AKI physiology, both associated with high mortality [64]. Single-arm studies in refractory ascites, monitoring for AKI, show mixed results. Hu 2024’s pilot study of empagliflozin reported one AKI case leading to withdrawal. In contrast, Kalambokis 2024 found no AKI incidence and a mild decrease in creatinine. Case series by Sharma 2023 and Seidita 2024 also noted mild creatinine decreases. However, these creatinine reductions at six months were not clinically significant as none of these patients had significant chronic kidney dysfunction (Estimated glomerular filtration rate (eGFR) < 60). Additionally, creatinine changes might also be due to lean body mass loss, not renal benefit. Interestingly, RCT data on AKI risk are conflicting. In the Singh 2024 study of dapagliflozin, AKI risk was significantly increased in the SGLT2I group. Conversely, in the Bakosh 2024 study on empagliflozin, the SoCD-alone group had a higher, though statistically insignificant, AKI risk. Comparing baseline characteristics reveals a key difference: liver disease severity, quantified by MELD-Na. The Singh 2024 study population had higher MELD-Na scores and showed increased AKI in the SGLT2I group. Therefore, it is possible that SGLT2I-induced volume depletion may be more likely to cause AKI in more critically ill patients. Furthermore, in Singh 2024, 12 of 13 AKI cases were attributed to sepsis. The SGLT2I group had a higher sepsis rate. It remains unclear if increased AKI was due to SGLT2I-induced volume depletion, sepsis, or a statistical anomaly from a small sample size. Therefore, in cirrhosis patients with higher MELD scores, SGLT2I use should warrant avoidance or careful renal monitoring.

### 4.6. Electrolyte and Acid-Base Derangements

Cirrhotic patients are susceptible to electrolyte disturbances for various reasons. Therefore, further electrolyte derangements from SGLT2Is added to SoCDs are concerning. These derangements can trigger arrhythmias or HE [25]. Bakosh 2024 reported significantly increased new-onset hyponatremia with SGLT2Is. However, Singh 2024 showed no difference in hyponatremia. Studies measuring pre- and post-sodium levels, including case series and prospective data, mostly observed mild increases or normalization of serum sodium [34,35]. One explanation is that in less severe cirrhosis or resolved ascites, reduced extracellular volume can elevate serum sodium. Conversely, patients with chronic, refractory ascites, often on maximal SoCD and multiple natriuretic agents, may experience excessive volume contraction. This could increase their risk of hypovolemic hyponatremia. Regarding potassium, SGLT2Is generally cause a mild lowering effect, but no major shifts in most populations [1]. Few included studies measured potassium levels. No clinically significant potassium changes or significant hypokalemia rates were found. Euglycemic Diabetic Ketoacidosis (DKA) is a rare, but severe and well-documented SGLT2I complication [1]. A single case series by Chao 2020 presented two cases of euglycemic DKA. These occurred in alcoholic cirrhosis patients with active alcohol use, developing 3 days and 3 weeks after SGLT2I initiation [32]. However, no other included studies reported any ketosis, including RCTs with substantial alcoholic cirrhosis populations. Nevertheless, adverse effect assessments in these studies excluded patients with active alcohol use. Therefore, more real-world evidence is needed to assess euglycemic DKA prevalence in this context. Until such evidence emerges, avoiding SGLT2Is in cirrhotic patients with active alcohol use is prudent.

### 4.7. Infection Risk

Cirrhotic patients are immunosuppressed due to the synthetic dysfunction of antibacterial proteins in both innate and adaptive immune systems [28]. This immunosuppression makes them susceptible to bacterial infections [29]. SGLT2Is promote glycosuria. It is therefore suggested that they might increase bacterial UTI risk [1]. General population data on SGLT2Is and UTIs show a significant increase in mycotic genital infections, but UTI incidence was not significantly increased until after 1 year [26]. Saffo’s 2020 single retrospective study in compensated cirrhosis supports this. Mycotic genitourinary infections were the most common adverse event, though bacterial UTIs were not observed. Regarding decompensated cirrhosis, Bakosh’s 2024 RCT showed a trend towards higher UTIs. This was not seen in Singh 2024, however, the SGLT2I group had 11 total infections versus four in the control group, which was statistically significant [35]. No other included study measured total infections. Literature on infection risk with SGLT2Is outside our review includes a retrospective case–control study. This study assessed bacteremia risk in urosepsis patients receiving SGLT2Is. Cirrhosis was the only comorbidity significantly associated with increased bacteremia risk in UTI patients [65]. Therefore, caution is advised before starting SGLT2Is in decompensated cirrhosis, especially those with additional immunocompromise or a history of urinary tract infection. Bacterial infection is a leading cause of acute-on-chronic liver failure, characterized by sudden hepatic decompensation, organ failure, and significantly increased mortality [66]. Therefore, more robust research is needed on UTI and other infection risks with SGLT2Is in this vulnerable population.

### 4.8. Potential Clinical Applicability

Data consistently demonstrates that SGLT2Is have been proven to be efficacious in augmenting SoCD or serving as an alternative diuretic when SoCDs are contraindicated in refractory ascites across various etiologies and regardless of T2DM status. However, further research is needed on the duration of this efficacy, as current studies span up to six months. Therefore, SGLT2Is’ most immediate role in refractory ascites may be palliative in patients with a life expectancy of less than 6 months. They could limit invasive procedures, especially given refractory ascites’ high one-year mortality (over 50%) [67]. Using SGLT2Is as first-line diuretics for ascites instead of SoCDs is unstudied and is likely inadvisable since SGLT2Is do not offer the same diuretic potency as furosemide [68]. In decompensated cirrhosis with non-refractory ascites, there is emerging evidence supporting SGLT2I use for ascites management, which is promising, but it needs to be confirmed with RCTs. Therefore, SGLT2Is should not currently be initiated for these specific indications. That said, SGLT2Is appear tolerated in patients with decompensated cirrhosis. Thus, continuation for other indications after decompensation resolution can be considered. This consideration should follow confirmation of safety in larger studies. Unsurprisingly, SGLT2Is offer similar hemodynamic benefits in cirrhosis as in heart failure. This similarity is due to shared pathophysiology, such as chronic sympathetic activity and third spacing [59]. Although some studies showed increased hypotension incidence, MAP reduction was short-lived. Furthermore, SGLT2Is improved diastolic dysfunction associated with cirrhotic cardiomyopathy [44]. These effects are consistent with benefits seen in HFpEF, suggesting potential cardiovascular advantages in cirrhotic patients as well. Therefore, continuation of SGLT2Is in cirrhosis patients with heart failure should be considered, when possible.

### 4.9. Limitations

Our review has a few limitations. In terms of the review process, cirrhosis was a keyword we focused on and, as a result, may have missed data on patients with MASLD that met the definition of cirrhosis via VCTE. The identified evidence varied in design. Of the studies with control groups, some compared SGLT2Is to SoCDs, whereas others compared SGLT2Is to other T2DM medications. Additionally, the severity and etiology of cirrhosis differed across studies and were limited to short-term durations up to 6 months for more severe forms of cirrhosis. This limited cross-study comparison and the meta-analysis we previously attempted [69] as each study’s results had to be considered in the context of the study design. The retrospective studies included were possibly prone to selection bias as cohorts exposed to SGLT2I likely represented more stable patients, so prospective data is needed to confirm SGLT2Is role in slowing the progression of liver disease. Finally, the safety data for SGLT2Is is limited to clinical trials where factors such as active alcohol use or concurrent acute illnesses were excluded, so more real-world data is necessary to assess their safety and to identify less common adverse effects.

## 5. Conclusions

The systematic review synthesizes current evidence on the role of Sodium–Glucose Cotransporter-2 Inhibitors (SGLT2Is) in managing liver cirrhosis. There was strong evidence to confirm their efficacy in the management of ascites by reducing large-volume paracentesis, particularly refractory ascites. Most data also suggested their role in slowing disease progression, although more RCTs are needed to measure changes in liver disease severity and rates of decompensation to confirm this. There is some data to suggest an increased risk of AKI and infection. Limitations include variability in control group comparisons, heterogeneous cirrhosis severity and etiology, potential selection bias in retrospective studies, and limited real-world safety data. Therefore, further research is needed to confirm SGLT2Is role in slowing disease progression and their safety in cirrhosis.

## Figures and Tables

**Figure 1 ijms-26-04781-f001:**
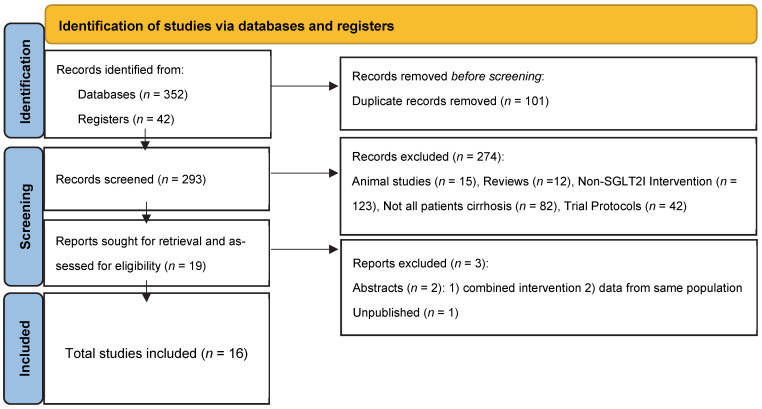
Preferred reporting items for systematic reviews and meta-analyses (PRISMA) 2020 flow diagram for systematic reviews, which included searches of databases and registers only.

**Figure 2 ijms-26-04781-f002:**
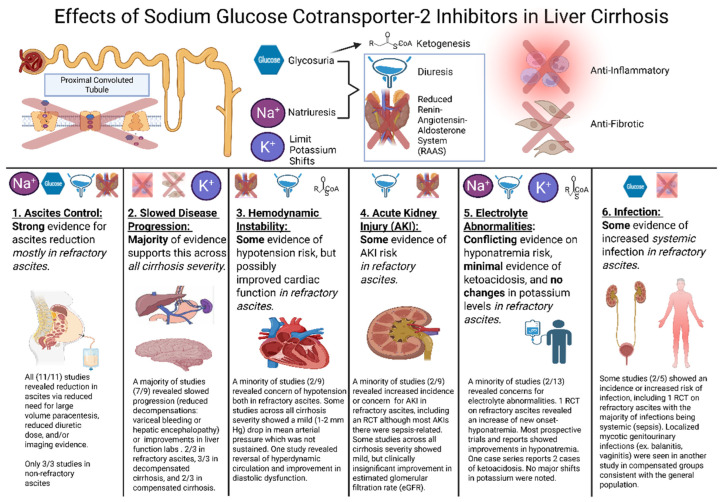
Overview of the Effects of Sodium Glucose Cotransporter-2 Inhibitors (SGLT2Is) in Liver Cirrhosis. The upper panel illustrates the key physiological mechanisms of SGLT2Is relevant to cirrhosis, including natriuresis and glycosuria, inducing diuresis, reduced renin–angiotensin–aldosterone system (RAAS) activity, and potential anti-inflammatory and anti-fibrotic effects. The lower panel summarizes our review findings on the clinical impact of SGLT2Is in liver cirrhosis across six key outcome domains tied to these mechanisms: ascites control, disease progression, hemodynamics, acute kidney injury risk, electrolyte abnormalities, and infection risk, highlighting the strength and consistency of the available evidence in each domain.

**Table 1 ijms-26-04781-t001:** Population characteristics, interventions, controls, primary or significant outcomes, and time periods (PICOT) are characteristics and major findings for comparative studies assessing SGLT2Is in liver cirrhosis.

Author Year	Design	SGLT2I (n)	Control (n)	% Decomp	% MASLD	MELD Na		Covariates	Inclusion:	Exclusion:	SGLT2I Effect on Primary Outcome	End Point (Weeks)
						Intervention	Control					
Saffo 2021 [37]	RC	NR 423	DPP4i 423	0	47	9.3	9.3	Age, Gender, MELD-Na, FIB-4, EV, CAD, HbA1c, medications EtOH use.	Compensated T2DM FIB-4 > 1.45	Prior diagnosis of ascites, History of TIPS, OLT	Ascites incidence insignificant decrease (HR: 0.68, CI: 0.37–1.25, =0.22)	144
Huynh 2023 [38]	RC	E 1403	MTF 1403	0	100	6.3	7.4	Age, Gender, Race/Ethnicity, Cirrhosis Etiology, FIB-4, EV MELD-Na, CAD, Hba1c, medications	Compensated T2DM	Prior decompensations Biliary Cirrhosis, T1DM	All-cause mortality significantly decreased (HR 0.57, CI: 0.41–0.81)	260
						*p* < 0.01						
Singh 2024 [35]	RCT	D 20	SoCD 20	100	20	22.0	22.0	Age, Gender, Cirrhosis Etiology, MELD-Na, Ascites Grade	Refractory ascites within 1 year	eGFR < 60, PVT, HCC, GIB, HE, Hypoglycemia, Hyponatremia, AKI, infection in past month	Significantly higher complete or partial control of ascites (70% vs. 35%, *p* = 0.04)	26
Bakosh 2024 [34]	RCT	E 21	SoCD 21	100	0	19.0	14.0	Age, Gender, Cirrhosis Etiology, MELD-Na/CP, Diabetes, Weight	Refractory ascites for >3 months	EtOH use, Hypotension, DKA	Significantly lower LVP need (42.9% vs. 100%, *p* < 0.01)	12
						*p* = 0.306						
Ayoub 2024 [39]	RC	NR 8038	SoCD 4019	100	NR	NR		Patient characteristics and comorbidities	Decompensated receiving SoCD	Alcoholic cirrhosis	Significantly lower rate of new decompensation at 1 month (28% vs. 39.2%, *p* < 0.001), 3 months (38% vs. 49.6%, *p* < 0.001), and 6 months (43.6% vs. 55.2%, *p* < 0.001) in the group receiving SGLT2I.	26
Seif El-Din 2024 [40]	PC	D 200	Insulin 100	73	24	14.4	17.5	Age, Gender, Cirrhosis Etiology CP	T2DM	Renal Impairment, Active Decompensation., Non-Insulin/SGLT2I DM treatments, T1DM	No primary outcome reported	12
						*p* = NR						

Six comparative studies (2021–2024) evaluating intervention of Sodium-glucose cotransporter-2 inhibitors (SGLT2Is) (NR = not reported, E = empagliflozin, D = Dapagliflozin) vs. control (MTF = Metformin, DPP4i = Dipeptidyl Peptidase-4 Inhibitors, SoCD = Standard of care diuretics) in patients with liver cirrhosis. Studies are characterized by design (RC = retrospective cohort, RCT = randomized controlled trial, PC = prospective cohort), sample sizes, demographics, and baseline disease severity measured by Model for End-Stage Liver Disease with Sodium (MELD-Na) scores. Primary outcomes show an overall trend toward clinical benefits with SGLT2Is, including improved ascites control, reduced need for large volume paracentesis (LVPs), decreased decompensation events, and reduced all-cause mortality in cirrhotic patients. Study durations ranged from 12 to 260 weeks, with varying inclusion and exclusion criteria primarily focused on compensated or refractory cirrhosis patients. (FIB-4 = Fibrosis-4 (FIB-4) Index for Liver Fibrosis, EV = Esophageal varices, CAD = Coronary artery disease, HbA1c = Hemoglobin A1c, EtOH = ethanol, T2DM = Type 2 Diabetes Mellitus, TIPS = Transjugular Intrahepatic Portosystemic Shunt, OLT = Orthotopic liver transplantation, HR = Hazard Ratio, CI = Confidence Interval, T1DM = Type 1 Diabetes Mellitus, eGFR = Estimated glomerular filtration rate, PVT = Portal Vein Thrombosis, HCC = Hepatocellular carcinoma, GIB = Gastrointestinal bleed, HE = Hepatic Encephalopathy, AKI = Acute Kidney Injury, CP = Child–Turcotte–Pugh Score, DKA = Diabetic ketoacidosis, MASLD = Metabolic-Associated Steatotic Liver Disease).

**Table 2 ijms-26-04781-t002:** Characteristics and major findings for non-comparative studies assessing SGLT2I in Liver Cirrhosis.

Author Year	Design	SGLT2I (n)	MASLD (%)	Decompensated (%)	Initial Condition	Clinically Significant Findings	End Point (Weeks)
Montalvo-Gordon 2020 [18]	CS	E (3)	100	100	Refractory ascites, with SoCD not tolerated, HE	Mean 7.5 kg weight loss associated with improved ascites control, normalization of hyponatremia	24
Saffo 2020 [41]	RCR	E (33) C (17) D (13) O (15)	50	19	Mainly compensated	10.2% developed NARHDs, 17% ascites, 3.8% liver mortality, 9.0% adverse events (85% mycotic genital infection), 1 patient had surgery for balanitis, no AKI or electrolyte disturbances	109
Chao 2020 [32]	CS	E(1) D(1)	0	0	Child Pugh A with chronic alcoholism for >10 years, Hba1c-8.4–8.5	Euglycemic DKA developed 3 days and 3 weeks after SGLT2I initiation	3
Kalambokis 2021 [17]	CR	E (1)	0	100	Hepatic Hydothorax not tolerating SoCD with recent HRS-AKI, Hyponatremia	Resolution of ascites and hyponatremia, no further clinical decompensations	16
Miyamoto 2021 [16]	CR	E (1)	100	100	6 rounds of CART for refractory ascites	Maintained ascites control off SoCD	12
Sharma 2023 [42]	PT	D (20)	85	0	All compensated patients with CSPH, mean CP~6.	3.4 kg weight loss	24
Shen 2024 [43]	PT	E (10)	10	50	Refractory ascites, with SoCD not tolerated (71%) or not working (29%)	Tolerated in decompensated cirrhosis with adverse events similar to heart failure and CKD data	4
Kalambokis 2024 [44]	PT	E (14)	NR	100	Refractory ascites, 50% had Na < 130, CP > 12	7 kg weight loss, marked increase in natriuresis, reduction in hyperdynamic circulation, reduction of RAAS activity.	12
Seidita 2024 [45]	CS (4)	D (4)	75	100	Severe abdominal ascites	75% ascites resolution and 2–3 pt. reduction in CP, normalization of mild hyponatremia	24
Hu 2024 [46]	PT	E (8)	50	100	Refractory ascites	Furosemide reduced from 80 mg to 40 mg	12

Ten non-comparative studies (2020–2024) evaluating Sodium-glucose cotransporter-2 inhibitors (SGLT2Is) (E = empagliflozin, D = Dapagliflozin, C = Canagliflozin, O = Other) in patients with liver cirrhosis. Studies are characterized by design (CS = case series, RCR = retrospective chart review, CR = case report, PT = pilot trial), with sample sizes ranging from single cases to 50 patients. Most studies focused on patients with refractory ascites or those not tolerating standard of care diuretics (SoCD). Primary outcomes demonstrate consistent trends of SGLT2I benefits, including improved ascites control, weight loss (3.4–7.5 kg), reduced diuretic requirements, normalization of hyponatremia, and enhanced natriuresis. Study durations ranged from 3 to 109 weeks, with most reporting favorable safety profiles despite the complex patient population. (NARHDs = Non-ascites related hepatic decompensations, CART = Cell-free and Concentrated Ascites Reinfusion Therapy, CSPH = Clinically significant portal hypertension, CP = Child–Turcotte–Pugh Score, RAAS = Renin–Angiotensin–Aldosterone System, HRS-AKI = Hepatorenal syndrome–acute kidney injury, MASLD = Metabolic-Associated Steatotic Liver Disease, HE = Hepatic Encephalopathy, AKI = Acute kidney injury, HbA1c = Hemoglobin A1c, DKA = Diabetic ketoacidosis, and CKD = Chronic Kidney Disease.

**Table 3 ijms-26-04781-t003:** Quality assessments.

Cochrane Risk of Bias 2.0											
Study	Randomization		Intervention Deviation		Missing Data		Outcome Measurement	Outcome Reporting		Total	Notes
Bakosh 2024 [34]	Low		Some		Some		Low	Low		Some Risk	Patients not blinded. 10% (*n* = 2) of SGLT2I group withdrew to AKI.
Singh 2024 [35]	Low		Low		Low		Low	Low		Low Risk	Low risk
**NIH Quality Assessment Tool for Case Series Studies**											
**Studies**	**Q1**	**Q2**	**Q3**	**Q4**	**Q5**	**Q6**	**Q7**	**Q8**	**Q9**	**Notes**	
Seidita 2024 [45]	Yes	Yes	Yes	Partially	Yes	Yes	Yes	No	Yes	Major limitations are small sample size and high drop-out rate in addition to limited statistical analysis and lack of comparison group. Heterogeneity in cirrhosis etiology and baseline characteristics.	
Saffo 2020 [41]	Yes	Yes	Yes	Partially	Yes	Yes	Yes	Yes	Yes	Retrospective, single center design, and lack of comparison group. Heterogeneity in etiology, disease severity, and treatments.	
Chao 2020 [32]	Yes	Yes	Yes	Yes	Yes	Yes	Yes	N/A	Yes	Small sample size, single center design, and lack of comparison group which limits generalizability	
Hu 2024 [46]	Yes	Yes	Yes	Partially	Yes	Yes	Yes	Yes	Yes	Small sample size, single center design, lack of comparison group, and relatively short follow-up period which limits generalizability. Heterogeneity in etiology and comorbidities	
Sharma 2023 [42]	Yes	Yes	Yes	Partially	Yes	Yes	Yes	Yes	Yes	Single center design, lack of comparison group, and high screen failure rate. This study only included compensated patients which may introduce selection bias. Heterogeneity in etiology and treatments which limits comparisons between patients.	
Montalvo-Gordon 2020 [18]	Yes	Yes	Yes	Partially	Yes	Yes	Yes	Yes	Yes	Small sample size (*n* = 2) and heterogeneity in etiology.	
Shen 2024 [43]	Yes	Yes	Yes	Yes	Yes	Yes	Yes	Yes	Yes	Small sample size (*n* = 10), relatively short follow-up, single-center study	
Kalambokis 2024 [44]	Yes	Yes	Yes	Yes	Yes	Yes	Yes	Yes	Yes	Small sample size (*n* = 14). Single-center study with no control group and open-label design.	
**ROBINS-I V2 for Observational Studies**											
**Studies**	**Confounding**		**Selection Bias**		**Classification**	**Deviation**	**Missing Data**	**Measurement of Outcome**	**Selection Reporting**	**Overall**	**Notes**
Huynh 2023 [38]	Moderate		Low		Low	Low	Low	Low	Low	Moderate	MELDNa higher in control group.
Saffo 2021 [37]	Low		Low		Low	Low	Low	Low	Low	Low	-
Seif El-Din 2024 [40]	High		Low		Low	Low	Low	Low	High	High	Suboptimal control group. No primary or secondary outcomes declared prospectively.
Ayoub 2024 [39]	Unclear		Moderate		High	Low	Low	Low	Low	High	Unable to classify indication for SGLT2I initiation and prevalence of refractory ascites. propensity-matched baseline characteristics not available.

Cochrane Risk of Bias 2.0 Assessment for Randomized Controlled Trials (RCTs). Quality assessment evaluated key domains of bias, including randomization process, intervention deviation, missing outcome data, outcome measurement, and outcome reporting. Each domain was rated as Low, Some, or High risk of bias. Both RCTs demonstrated generally good methodological quality with predominantly low risk of bias across domains, though Bakosh 2024 had some concerns regarding intervention deviation. (SGLT2I = Sodium-glucose cotransporter-2 inhibitors, AKI = Acute Kidney Injury) The tool can be accessed at: https://methods.cochrane.org/risk-bias-2 (accessed on 14 February 2025). NIH Quality Assessment Tool Evaluation of Case Series Studies. Quality assessment was performed across nine domains (Q1–Q9), evaluating study objectives, population characteristics, intervention implementation, outcome measurement, and statistical analysis. While most studies met basic quality criteria, common limitations included small sample sizes, lack of comparison groups, and heterogeneity in baseline characteristics and disease etiology. ‘Yes’ indicates the criterion was met, ‘Partially’ indicates incomplete fulfillment, and ‘N/A’ indicates the criterion was not applicable. The tool can be accessed at: https://www.nhlbi.nih.gov/health-topics/study-quality-assessment-tools (accessed on 14 February 2025). Risk Of Bias in Non-randomized Studies of Interventions (ROBINS-I) V2 Assessment. Evaluation included assessment of confounding, participant selection, intervention classification, deviation from intended interventions, missing data, outcome measurement, and selection of reported results. Studies demonstrated varying quality, with key limitations in confounding control and selection bias. Each domain was rated as Low, Moderate, High, or Unclear risk of bias, with an overall risk assessment provided. (SGLT2I = Sodium-glucose cotransporter-2 inhibitors, MELDNa = Model for End-Stage Liver Disease with Sodium.) The tool can be accessed at: https://www.riskofbias.info/welcome/robins-i-v2 (accessed on 14 February 2025).

**Table 4 ijms-26-04781-t004:** Reported outcomes.

Outcome	Associated with SGLT2I	No Difference (No Incidence in Single-Arm)	Associated with Control (or Opposite Effect in Single Arm)
Ascites Reduction	Refractory: Singh 2024 [35], Bakosh 2024 [34], Montalvo-Gordon 2020 [18], Kalambokis 2021 [17], Miyamoto 2021 [16], Shen [43], Kalambokis 2024 [44], Hu 2024 [46] Decompensated: Seif El-Din 2024 [40], Ayoub 2024 [39], Seidita 2024 [45]		
Slowed Disease Progression	Refractory: Bakosh 2024 [34], Kalambokis 2021 [17] Decompensated: Seidita 2024 [45], Seif El Din 2024 [40], Ayoub 2024 [39] Compensated: Huynh 2023 [38], Sharma 2023 [42]	Refractory: Singh 2024 [35] Compensated: Saffo 2021 [37]	
Hemodynamic Instability	Refractory: Shen 2024 [43], Hu 2024 [46]	Refractory: Singh 2024 [35], Bakosh 2024 [34], Miyamoto 2021 [16], Kalambokis 2021 [17] Decompensated: Seif El-Din 2024 [40] Compensated: Sharma 2023 [42]	Refractory: Kalambokis 2024 [44]
AKI/HRS Risk	Refractory: Singh 2024 [35], Hu 2024 [46]	Refractory: Bakosh 2024 [34], Kalambokis 2021 [17], Miyamoto 2021 [16], Shen 2024 [43], Kalambokis 2024 [44] Compensated: Saffo 2020 [41], Sharma 2023 [42]	
Electrolyte/Acid Base abnormalities (Hyponatremia, Hypokalemia, Ketoacidosis)	Refractory: Bakosh 2024 [34], Chao 2020 [32]	Refractory: Singh 2024 [35], Hu 2024 [46] Decompensated: Seif El Din 2024 [40] Compensated: Saffo 2020 [41]	Refractory: Montalvo-Gordon 2021 [18], Kalambokis 2021 [17], Miyamoto 2021 [17], Kalambokis 2024 [44], Shen [43] Decompensated: Seidita 2024 [45] Compensated: Sharma 2023 [42]
Infection Risk (all infections)	Refractory: Singh 2024 [35] Compensated: Saffo 2020 [41]	Refractory: Bakosh 2024 [34], Kalambokis 2024 [44]	Decompensated: Seif-El Din 2024 [40]

Key clinical outcomes across studies examining sodium-glucose cotransporter-2 inhibitors (SGLT2Is) in liver cirrhosis, organized by effect association. For ascites reduction, multiple studies reported benefits with SGLT2I therapy in both refractory and decompensated settings, with no studies showing neutral or negative effects. Liver function benefits were demonstrated in most studies across all cirrhosis stages, though Saffo 2021 and Singh 2024 reported no differences in certain patient populations. Hemodynamic instability risks were minimal, with most studies reporting no difference in incidence, while only Shen 2024 and Hu 2024 noted hemodynamic concerns in refractory ascites patients. (AKI = Acute Kidney Injury, HRS = Hepatorenal syndrome).

## Data Availability

All data files were extracted from tables, figures, or texts available in the main cited articles or Appendix A. No novel datasets were generated or analyzed in this manuscript.

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
