# Peer review of "Sodium-Glucose Cotransporter-2 Inhibitors in Liver Cirrhosis: A Systematic Review of Their Role in Ascites Management, Slowing Disease Progression, and Safety"

_ijms, 2025, doi:10.3390/ijms26104781_

Round 1
Reviewer 1 Report
Comments and Suggestions for Authors
The review article titled (The Effects of Sodium-Glucose Cotransporter Protein-2 Inhibitors in Liver Cirrhosis: An Updated Systematic Review of the Literature) by Dhoop et al. discussed the role of The pharmacological inhibitiors of Sodium-Glucose Cotransporter Protein-2 in treating Liver Cirrhosis in a Systematic Review of the Literature. This is a useful review in which the authors summarized a novel point and I have the following recommendations to improve the final shape of the review.
1 - The title (The Effects of Sodium-Glucose Cotransporter Protein-2 Inhibitors in Liver Cirrhosis: An Updated Systematic Review of the Literature) needs some revisions: please remove (The) and mention what was the proposed effect? was it protective or curative
2 - ALso give an idea about the nature of the reviewed effect, was it curative or preventive?
3 - Abstract: is short and needs some amendments with numerical values if possible
4 - Key words : authors must add key words
5 - In introduction : Line 31, authors wrote (They were approved by the Federal Drug Administration (FDA) for glycemic control in Type 2 Diabetes Mellitus (T2DM)[1]
Please mention the date of this approval
6 - In introduction : please mention the reported adverse effect of the Sodium-Glucose Cotransporter Protein-2 Inhibitors other than those mentioned in line 61 - 68
7 - In abstract: authors wrote and concluded that ( Limited, lower- 23
quality data suggest possible hepatic benefits in compensated and decompensated cirrho- 24
sis. Larger studies over longer periods across various etiologies and severities of liver cir- 25
rhosis assessing ascites control and safety are needed as well as higher quality data as- 26
sessing whether SGTL2Is improve hepatic function and decrease liver-related events.)
So, I think authors should give a stronger rationale why they decided to write this review while limited data are available for these new drugs
8 - Introdcution line 81: authors wrote (herefore, the pur- 81
pose of this review is not to provide precise data on the efficacy or safety of SGLT2Is for 82
any particular indication in cirrhosis, but rather to systematically tabulate and appraise 83
the various studies on SGLT2Is in cirrhosis to identify trends and gaps in the literature to 84
drive future investigation.)
Please ensure that all these aims have been covered in the conclusion of the review
9 - please write the limitations of the review
10 - Table 2 line 282: the Footnote is not correct (please remove Table 2 ) from the footnote
Also all the tables
11 - Authors need to explroe this to better describe the rationale and novelty of the study.
12 - Ensure every abbreviation is explained at the first appearnace in abstract & then in the body text
13 - Every abbreviation in figures should be explained in the figure legend to be self explanatory & stands alone.
14 - This nice topic deserves to draw a diagram to illustrate the main points reviewed in the manuscript. This will facilitate reading and comprehension of the article.
Author Response
Below, please find point-by-point responses to the reviewers recommendations.
Title/Abstract:
1 - The title (The Effects of Sodium-Glucose Cotransporter Protein-2 Inhibitors in Liver Cirrhosis: An Updated Systematic Review of the Literature) needs some revisions: please remove (The) and mention what was the proposed effect? was it protective or curative
2 - Also give an idea about the nature of the reviewed effect, was it curative or preventive?
To address the first point, we agree with the reviewers’ point regarding the title. Our review does not focus on a specific effect, but we agree effects is a generic phrase. To address the second point, we divided the effects into 3 points: Ascites Management, Slowing Disease (Cirrhosis) Progression, and Safety (line 1-4). This is later expanded into 6 domains.
3 - Abstract: Is short and needs some amendments with numerical values if possible
Initially, we recorded the number of studies tallied for just the benefits (reduction in ascites and improved hepatic function). We have now added all 6 outcome domains (line 20-22) and given tallies for each domain (lines 24-29). This gives readers a sense of the number of studies out of total supporting each domain. Additionally, we have placed transition of “appraisal of the literature” (line 29-31) to convey that our conclusion is not strictly based on a numerical tally, but rather appraisal of the studies in the as outlined by the quality assessment and in prose throughout the discussion. Abstract was originally 166 words, it is now 212 words (approximately 200 words maximum) with addition of this information.
4 - Key words : authors must add key words
We have added 10 keywords to reflect the 6 domains we assessed (line 38-39)
Introduction:
5 - In introduction: Line 31, authors wrote (They were approved by the Federal Drug Administration (FDA) for glycemic control in Type 2 Diabetes Mellitus (T2DM)[1] Please mention the date of this approval.
We have added the year 2013 (line 38) which adds a timeline of 2013->2015-2020->2024 when most of these newer studies which added consistency.
6 - In introduction : please mention the reported adverse effect of the Sodium-Glucose Cotransporter Protein-2 Inhibitors other than those mentioned in line 61 – 68
General overview of adverse effects of SGLT2Is would serve as valuable context so readers can assess which factors are relevant to cirrhosis. This was been added to line 73-76.
7 - In abstract: authors wrote and concluded that ( Limited, lower- 23
quality data suggest possible hepatic benefits in compensated and decompensated cirrho- 24
sis. Larger studies over longer periods across various etiologies and severities of liver cir- 25
rhosis assessing ascites control and safety are needed as well as higher quality data as- 26
sessing whether SGTL2Is improve hepatic function and decrease liver-related events.) So, I think authors should give a stronger rationale why they decided to write this review while limited data are available for these new drugs
This feedback was very valuable in improving the overall impression of the manuscript. To summarize, we feel that excessively critical language was used to describe the studies (i.e. higher quality data needed) was distracting and rightfully may make readers question the rationale for the review. We revisited the abstract. We draw a distinction between what the current evidence strongly supports vs. potentially suggestions (line 26-28)
- We agree rationale for the study needs more emphasis. It is now explicitly stated in the abstract (line 14-16) and we have expanded on this point in the introduction (lines 44-51).
- We added an additional study (reference 12) which demonstrated SGLT2I was associated with reduced incidence of esophageal varices, a major source of mortality in cirrhosis although this study was not included in the studies of the systematic review as not all patients had cirrhosis. (lines 59-61). This further justifies the importance of this topic.
8 - Introduction line 81: authors wrote (herefore, the pur- 81
pose of this review is not to provide precise data on the efficacy or safety of SGLT2Is for 82
any particular indication in cirrhosis, but rather to systematically tabulate and appraise 83
the various studies on SGLT2Is in cirrhosis to identify trends and gaps in the literature to 84
drive future investigation.)
As stated above, we have limited excessively critical language but have still included gaps in the literature to drive further investigation. Therefore, we have rewritten our purpose statement in the title (lines-1-4) introduction (lines 97-99). We maintained the portion regarding gaps (lines 98-99). These gaps are introduced in the abstract (lines 28-30), expanded on in limitations (lines 563-571), and reiterated in the conclusion (lines 581-582).
9 - Please write the limitations of the review
Limitations are outlined as a stand-alone section (562-572). The main limitations are listed in the conclusion (lines 580-583) for consistency.
10 - Table 2 line 282: the Footnote is not correct (please remove Table 2 ) from the footnote. Also all the tables
Footnotes were redundant and have been removed.
11 - Authors need to explore this to better describe the rationale and novelty of the study.
Please see points made in point 7 to explain the rationale of the studies. Regarding the novelty, currently, there is an abundance of new literature on the topic with 8 novel studies including the first 2 randomized controlled trials that have not been synthesized into a previous major review of SGLT2Is on this topic so therefore this topic deserves an updated review. This was expanded on in lines 93-96.
12 & 13 - Ensure every abbreviation is explained at the first appearnace in abstract & then in the body text. Every abbreviation in figures should be explained in the figure legend to be self-explanatory & stand alone.
Abbreviations are explained at first appearance and every abbreviation in figure was explained again in figure legend to stand alone
14 - This nice topic deserves to draw a diagram to illustrate the main points reviewed in the manuscript. This will facilitate reading and comprehension of the article.
We have generated a diagram to illustrate the main points of the article listed as Figure 2. This summarizes the main findings of the 6 domains
Created in BioRender. Dhoop, S. (2025) https://BioRender.com/72gabnt with license to publish.

Reviewer 2 Report
Comments and Suggestions for Authors
Congratulations to their review for the relevance of their manuscript; I have some comments about it:
The authors are encouraged to incorporate, where available, recently published studies on the use of SGLT2 inhibitors in heart failure with preserved ejection fraction and their potential relevance to cirrhotic cardiomyopathy and emerging safety data on SGLT2I-associated acute kidney injury stratified by MELD-Na scores from large-scale population cohorts; as well as real-world registry data (e.g., from Optum or the VA) that could offer broader insights into outcomes such as transplant-free survival, hospital readmission for hepatic encephalopathy, and progression of cirrhotic cardiomyopathy.
The manuscript would benefit from a dedicated section or paragraph addressing the pleiotropic effects of SGLT2 inhibitors beyond glycemic control and natriuresis. These agents have demonstrated anti-inflammatory, antifibrotic, antiarrhythmic, and metabolic benefits; in order to empower their manuscripts authors should include and briefly discuss these new evidences (doi: 10.1002/ehf2.15223.)
Additionally, the inclusion of a visual summary—such as a schematic "mechanism of action" box—depicting the proposed physiological pathways of SGLT2Is in cirrhosis (e.g., natriuresis, modulation of mean arterial pressure, anti-inflammatory effects) would enhance the clarity and educational value of the manuscript.
Author Response
Congratulations to their review for the relevance of their manuscript; I have some comments about it:
1. The authors are encouraged to incorporate, where available, recently published studies on the use of SGLT2 inhibitors in heart failure with preserved ejection fraction and their potential relevance to cirrhotic cardiomyopathy and emerging safety data on SGLT2I-associated acute kidney injury stratified by MELD-Na scores from large-scale population cohorts; as well as real-world registry data (e.g., from Optum or the VA) that could offer broader insights into outcomes such as transplant-free survival, hospital readmission for hepatic encephalopathy, and progression of cirrhotic cardiomyopathy.
Despite the aid of our experienced librarian, we can say with high level of certainty that no studies stratifying SGLT2I-associated AKI by MELD-Na or other liver severity are published at this time although our group will be looking to publish such data in the future. However, we did make a few new points here:
1. We have added data on SGLT2I inhibitors in HFpEF and their potential relevance to cirrhotic cardiomyopathy in lines 472-481.
2. Additionally, we have expanded on the discussion regarding hepatic encephalopathy in the slowed disease progression section (lines 441-442)
2. The manuscript would benefit from a dedicated section or paragraph addressing the pleiotropic effects of SGLT2 inhibitors beyond glycemic control and natriuresis. These agents have demonstrated anti-inflammatory, antifibrotic, antiarrhythmic, and metabolic benefits; in order to empower their manuscripts authors should include and briefly discuss these new evidences (doi: 10.1002/ehf2.15223.)
3. We reviewed the literature. While the suggested citation and antiarrhythmic properties were outside the scope of our review, we did find a 2025 meta-analysis revealing a potential anti-neoplastic role for SGLT2Is. We briefly discuss this new evidence to provide further mechanistic evidence that SGLT2Is may slow progression of cirrhosis and also suggest mechanisms for stated findings of our review (lines 435-449, Figure 2).
3.) Additionally, the inclusion of a visual summary—such as a schematic "mechanism of action" box—depicting the proposed physiological pathways of SGLT2Is in cirrhosis (e.g., natriuresis, modulation of mean arterial pressure, anti-inflammatory effects) would enhance the clarity and educational value of the manuscript.
We have generated a diagram to illustrate the main points of the article listed as Figure 2. This summarizes the main findings of the 6 domains and ties them to the mechanisms of action that we propose to be responsible for each finding. We feel it is a quick summary to enhance the educational value of the manuscript.

Round 2
Reviewer 1 Report
Comments and Suggestions for Authors
The revised version of review titled (The Effects of Sodium-Glucose Cotransporter Protein-2 Inhibitors in Liver Cirrhosis: An Updated Systematic Review of the Literature) by Sudheer Dhoop et al. was improved compared to the original submission.
I am pleased to recommend its acceptance in IJMS
Reviewer 2 Report
Comments and Suggestions for Authors
Congratulations to the authors for the revised version of their manuscript.